# Synthesis and Configurational Character Study of Novel Structural Isomers Based on Pyrene–Imidazole

**DOI:** 10.3390/molecules24122293

**Published:** 2019-06-20

**Authors:** Yu-Long Liu, Liu Yang, You-Quan Guo, Guo-Qiang Xu, Bin Qu, Ying Fu

**Affiliations:** 1Department of Applied Chemistry, College of Science, Northeast Agricultural University, Harbin 150030, China; liuyulong@neau.edu.cn (Y.-L.L.); yangliu@neau.edu.cn (L.Y.); xuguoqiang@neau.edu.cn (G.-Q.X.); qubin@neau.edu.cn (B.Q.); 2CAS Key Laboratory of Renewable Energy, Guangzhou Institute of Energy Conversion, Guangzhou 510640, China; 3Heilongjiang Vocational College of Biology Science and Technology, Harbin 150025, China; feilong50123146@126.com

**Keywords:** one-pot reaction, pyrene–imidazole, structural isomer, molecular symmetry, crystal structure

## Abstract

Isomers provide more possibilities for the structure of organic compounds. Molecular structures determine their corresponding properties, therefore the intrinsic relationship between structure and properties of isomers is of great research value. Isomers with a stable structure and excellent performance possess more potential for development and application. In this paper, we design and synthesize structural isomers with different molecular symmetries based on the asymmetric structure of imidazole and the symmetrical structure of pyrene. Isomers with stable molecular structures can be obtained by a simple and efficient “one-pot” reaction, involving axisymmetric configuration and centrosymmetric configuration. Using this “click-like” reaction, the structure of target molecules is controllable and adjustable. Furthermore, the effect of molecular configurations on molecular stacking of crystal is studied. The variation of the optical and thermal properties, the optimized structures, and orbital distributions of isomers depends on different molecular geometry with different symmetry, which are revealed by crystallographic analysis. This present strategy provides an efficient synthetic method for the design and synthesis of structural isomers based on pyrene–imidazole.

## 1. Introduction

Isomers are one of the major research topics in chemistry and materials. In the field of medicinal and material science, the pharmacological properties and the macroscopic properties of isomers may be different or even dramatically distinctive [1,2,3,4]. In the field of organic optoelectronics, isomers have also drawn a lot of attention in recent years [5]. Structural isomers, for instance, which possess the same bonding mode but hold different geometric positions of related atoms in space, may not keep similar properties necessarily, even though they have the same functional groups; therefore, these valuable isomers can be provided as model compounds to research the relationship between structure and properties [6,7]. Besides, structures of structural isomers are stabilized through chemical bonds where photochemical or thermochemical isomerization could not occur suggesting excellent stability of structural isomers. A great deal of the intensive study toward the *syn*-isomeric and *anti*-isomeric effects on the crucial changes of photophysical properties for thiophene derivatives has taken place and thus becomes typical [8,9,10,11]. It has been shown that much higher charge mobilities appear in pure isomers with *anti*-configuration than their corresponding *syn*-counterparts in the performance of organic field effect transistors, and this result has been reported by several groups involved with the model compounds of anthradithiophene, dicyanomethylenedithiophene, anthrabisbenzothiophene, and other systems [9,10,11]. In addition, emitters with naphthoimidazole (NI) unit were first reported by Su’s group [12]. Combined with triphenylamine, a series of donor-accepter molecules based on naphtho[1,2-d]imidazole structural isomers were synthesized, and their different configuration effects on the performance of non-doped single-layer fluorescent organic light-emitting diodes were systematically studied. 1*H*-naphtho[1,2-d]imidazole-based compounds emanate similar deep-blue light emission to 3*H*-naphtho[1,2-d]imidazole analogues but show much higher efficiency. The author pointed out that the best performance in single-layer devices of 3*H*-naphtho[1,2-d]imidazole compounds is attributed to their much higher capacity for direct electron-injection from the cathode. Moreover, structural isomers could be designed and synthesized through regulating the reactive sites thus providing tremendous examples of structure–property study [13,14,15,16,17]. These valuable studies have promoted the vigorous development of organic functional materials in organic electronics and made great contributions to structure–property relationship of structural isomers in organic chemistry.

Exploiting structure and potential of novel isomers is a never-ending task for a scientist. Pyrene, a classical polycyclic aromatic hydrocarbon which consists of four fused benzene rings, represents one of the mostly investigated organic molecules over the past decades [18]. The planar structural motif, large π-system, and good mobility of π-electron flow endow pyrene with attractive photophysical properties and have been thoroughly investigated according to admirable fluorescent and self-assembling property in recent years [19,20,21]. Nowadays, there has also been increasing interest in utilizing the pyrene unit as a semiconductor in the optoelectronic field [22,23,24]. It is widely known that the molecular structure and supramolecular structure of materials can be regulated through chemical modification at different sites of molecules, thereby affecting their relevant performance on applications. Therefore, efficient synthesis and novel structure of pyrene-based materials is of great interest. 1,3-Imidazole belongs to a five-numbered-ring functional building block and is commonly used for efficient light emitting materials such as benzimidazole, naphtho[1,2-d]imidazole, and phenanthro[9,10-d]imidazole in virtue of its amphoteric characteristics originated from the two different kinds of sp^2^ hybrid nitrogen atoms. The asymmetric structure of imidazole provides a priori condition for the construction of structural isomers. Based on these considerations, a convenient and one-pot facile methodology is used to prepare a novel polycyclic skeleton straightforwardly using imidazole and pyrene as the key units. Although many efforts have been made to prepare pyrene–imidazole derivatives with different substitutions at one side of the K-region on pyrene [25,26,27], there is rarely an attempt to synthesize a bilateral pyrene–imidazole-based polycyclic skeleton in which the K-region of pyrene is fully modified [28,29,30].

In this study, small organic molecules with a clear structure and structural isomers with different symmetries are selected as the research object. With this consideration in mind, novel structural isomers with *syn*-/*anti*-configuration consisting of a rigid π-structure of pyrene and nonsymmetrical unit of imidazole are obtained through a one-pot reaction. In accordance with previous research [31,32,33], the “click-like” one-pot reaction is a modification of the general method of the Debus–Radziszewski reaction which was presented more than 100 years ago by Debus and Radziszewski and has proved to be a highly efficient multicomponent reaction (MCR) yielding imidazole under mild conditions [34,35]. The one-step procedure utilized in this paper is simple, efficient, and structurally controllable whereby ammonium acetate, arylamine, pyrene-4,5,9,10-tetraones, and aromatic aldehyde in the solution of acetic acid are treated by reflux for 2 h. Eventually centrosymmetric *anti*-tbu-PyDPI and axisymmetric *syn*-tbu-PyDPI are readily and simultaneously generated during the synthesis process (Scheme 1). The two structural isomers are separated successfully with high purity. Furthermore, the structures of *anti*-tbu-PyDPI with centrosymmetry and *syn*-tbu-PyDPI with axisymmetry were both determined by X-ray single-crystal diffraction precisely.

## 2. Results and Discussion

Synthesis route of the *anti*/*syn*-isomers of tbu-PyDPI is depicted in Scheme 2. *Anti*-tbu-PyDPI and *syn*-tbu-PyDPI were synthesized through a one-pot reaction simultaneously with a moderate yield of 35.0% and 32.5%, respectively. When separating and purifying by chromatography, the *R_f_* values of these two isomers varied significantly after being eluted with a mixed solvent of petroleum ether and dichloromethane in a ratio of 1:5 *v*/*v*. This was a consequence of the differential molecular polarities of the *syn*- and *anti*-isomers with various symmetries, leading to noticeable differences in the molecular shape and dipole moment of these rigid polycyclic aromatic hydrocarbons. Thus, *anti*- and *syn*-tbu-PyDPI could be obtained successfully in an approximate 1:1 ratio.

### 2.1. ^1^H NMR Spectrum Identification

As shown in the ^1^H NMR spectra (Figure 1), the resonance peaks of protons H_a_ of tertiary butyl on 2 and 7-position of pyrene in *anti*-tbu-PyDPI occurred at 1.31 ppm exclusively, demonstrating that they had the same chemical atmosphere. As a result of centrosymmetric configuration, the shielding effect of protons H_a_ was merely from one phenyl ring attached to the N atom of imidazole. In contrast, those relevant resonance peaks of protons in *syn*-tbu-PyDPI, marked as H_b_ and H_c_, displayed a relatively high-field chemical shift of 0.84 ppm and relatively low-field chemical shift of 1.69 ppm, respectively. This difference regarding the chemical shift of protons on tertiary butyl could be derived from the fact that the ring-current effect [36], in which two phenyl rings connecting to the N atom of imidazole resulted in a relatively strong shielding effect toward protons H_b_, gaving rise to a concomitant upfield chemical shift (0.84 ppm). Meanwhile, there is the weakest shielding effect of H_c_ leading to a concomitant downfield chemical shift (1.69 ppm). Consequently, the chemical shift of H_a_ was located at the middle (1.31 ppm) between that of H_b_ (0.84 ppm) and H_c_ (1.69 ppm). The characteristic ^1^H NMR resonance of protons for these two isomers remained unchanged after ultraviolet irradiation or heating to 120 °C. This suggests that photochemical or thermocatalytic isomerization could not appear in *syn*/*anti*-tbu-PyDPIs indicating the excellent stability of these structural isomers.

### 2.2. Crystal Structure

Single crystals of two isomers were grown by sublimation and the single-crystal measurement further confirmed the configuration of *syn*/*anti*-tbu-PyDPI precisely. Their crystal structure with atomic numbering is given in Figure 2. The numbering guide indicates a configuration of centrosymmetry for *anti*-tbu-PyDPI where the phenyl planes (C(1)–C(2)–C(3) and C(1A)–C(2A)–C(3A)) make the same angle of 34° with the imidazole plane and the other phenyl planes (C(8)–C(9)–C(10) and C(8A)–C(9A)–C(10A)) also make the same angle of 80° with the imidazole plane (Figure 2 and Figure 3a). In contrast, all atomic numbering varied for *syn*-tbu-PyDPI reflecting its axisymmetry with a slightly different torsion angle between the phenyl plane and imidazole plane (Figure 3b).

The derived crystal structures are shown in Figure 3 with selected molecular parameters provided in Appendix A. As depicted in Figure 3, both crystals crystallized in a monoclinic space group *P* 2_1_/*c*. The *anti*/*syn*-tbu-PyDPI crystals had an essentially planar core and pyrene–imidazole ring, while their formed molecular columns packed in a herringbone fashion for both of them. The *anti*-crystal contained only two molecules in the unit cell while there are four molecules in the *syn*-crystal which exhibited a more ordered arrangement of the *anti*-part. Due to the axisymmetric configuration of *syn*-tbu-PyDPI which can create an induced dipole along the short axis direction (Figure 4), it is proposed that formation of alternating the antiparallel arrangement in a flat dimer of one unit cell was driven by the dipole–dipole interactions [37], referring to the pyrene–imidazole unit with an interfacial distance of ca. 4.89 Å. Because of the steric hindrance effect of tert-butyl, the adjacent molecules adopt a slip-stacked stacking mode along the short molecular axis and the long molecular axis synchronously to some extent (Figure 4).

We further analyzed the intermolecular interactions for *anti*/*syn*-crystal in unit cell. There exist many various kinds of C-H•••π interactions between two adjacent molecules in one unit cell (Appendix A). These intermolecular CH(sp^3^/sp^2^)•••π interactions are the key forces of the stacking mode. The CH•••π hydrogen bond in organic crystal is one of the most important weak intermolecular forces, significant in controlling molecular configurations and playing a vital role of various natures in chemistry [38]. In spite of the stabilization coming from a single intermolecular CH•••π interaction being weak, the multicomponents of intermolecular CH•••π interactions can be of great importance. As shown in Appendix A, in *anti*-isomer, it is C7···C13 (–x, –y, –z) of 3.44 Å (C7···H13(–x, –y, –z) of 2.63 Å), while in *syn*-isomer, it is C20···C42 (1–x, –y, 1–z) of 3.65 Å (C20···H42(1–x, –y, 1–z) of 2.73 Å). Apparently, the molecule stacking of alternating the antiparallel alignment of the pyrene–imidazole unit in the *syn*-crystal is attributed to the van der Waals force derived from a weak intermolecular dipole (Figure 4). However, no intermolecular π–π interaction upon face-to-face between pyrene units is perceived for these structural isomers. Besides, the type and number of CH•••π interaction may influence the photophysical properties of materials, especially for organic molecules containing a π-system [39].

### 2.3. Optical Properties

The normalized UV-vis absorption spectra and photoluminescence (PL) spectra of the isomers in dilute THF with sample concentrations of 10^−5^ M are shown in Figure 5. They manifested similar absorption and PL spectra due to their corresponding molecular organization. The absorption spectra of the two molecules displayed complex absorption profiles dominated by a series of bands resulting from the multiple localized π–π* transitions which originate from phenyl and the pyrene–imidazole moiety [25,28,29]. The molar extinction coefficient of the peak at 304 nm for *anti*-tbu-PyDPI and the peak at 314 nm for *syn*-tbu-PyDPI is 8.2 × 10^4^ M^−1^ cm^−1^ and 6.3 × 10^4^ M^−1^ cm^−1^ respectively, which is large for a π–π* transition character. Moreover, the main absorption peak of *syn*-tbu-PyDPI (314 nm) is slightly red-shifted (10 nm) compared with that of *anti*-tbu-PyDPI (304 nm) (Figure 5) which may be attributed to the differentia in molecular configuration and symmetry. There is no absorption at the low energy region demonstrating that no intramolecular charge transfer exist which further confirms their symmetric molecular configurations. The PL spectrum of *anti*-tbu-PyDPI in dilute THF shows three distinctive bands located at 392, 414, and 438 nm. The emission spectrum of *syn*-tbu-PyDPI shows the same shape with a slight redshift of approximately 2 nm whose bands are located at 394, 416, and 440 nm. Their PL spectra indicated the conformation of fluorophore of isomers from the ground to the first excited singlet states in solutions were nearly the same which could be attributed to the pyrene–imidazole group. The deep-blue emission with obvious vibronic structure features illustrates the blue-fluorescence property of pyrene–imidazole based derivatives [26,27,28].

### 2.4. Density Functional Theory Calculation

Density functional theory (DFT) calculations with a B3LYP/6-31G(d,p) basis set using the Gaussian 03W program were performed to further address their differentia in symmetry. The optimized molecular configuration is shown in Figure 6. The pyrene–imidazole segment presented nearly coplanar ground state geometries. The dihedral angles between the phenyl ring attached to the N atom of imidazole and imidazole ring of *anti*-isomer and *syn*-isomer are close to 80°. The twisty molecular structure may impede the occurrence of intermolecular aggregation to some extent. The dihedral angles between the phenyl ring attached to the C atom of imidazole and imidazole ring of *anti*-isomer and *syn*-isomer are 30° and 31°, respectively. These calculations are basically consistent with the results of monocrystalline structures mentioned above. In addition, the distributions of the front molecular orbitals are shown in Figure 7. The highest occupied molecular orbitals (HOMOs) of *anti*-tbu-PyDPI and *syn*-tbu-PyDPI were both delocalized over the molecule along the long molecular axis. Their lowest unoccupied molecular orbitals (LUMOs) were delocalized over the whole molecule, while there are more distributions on the bottom part of pyrene close to the two phenyl rings attached to the N atom of imidazole for *syn*-tbu-PyDPI which probably resulted from the distinct dipole orientation in *syn*-tbu-PyDPI (Figure 7). In general, the HOMO and LUMO mainly delocalized over the pyrene–imidazole segment, indicating a typical pure π→π* transition for both of them, which can also be confirmed from their PL spectra (Figure 5). Meanwhile, the distribution of LUMOs of *syn*-isomer further prove its axisymmetric configuration whereby a certain intramolecular dipole can be formed in the direction of molecule’s short axis [29].

### 2.5. Thermal Analysis: The DSC and TGA Graphs of Syn/Anti-tbu-PyDPI

Such a dual five-membered-ring skeleton demonstrated high thermal stability. From their thermogravimetric analysis (TGA) graphs, *anti*-tbu-PyDPI and *syn*-tbu-PyDPI exhibited outstanding thermal stability with decomposition temperature (*T*_d_) (5% weight loss) as high as 520 and 500 °C, respectively (Figure 8). The single pyrene–imidazole unit already revealed high thermal stability [28,29], and the centrosymmetric structure of *anti*-tbu-PyDPI further increased the thermal stability of the material. In differential scanning calorimetry (DSC) heating cycles up to 450 °C, *anti*-tbu-PyDPI did not show any thermal transition, which was due to its highly rigid structures and multi-intermolecular interactions (Appendix A). *T*_m_ appeared at 420 °C for *syn*-tbu-PyDPI, which reflects that an axisymmetric structure shows better molecular flexibility than a centrosymmetric structure due to the comparatively lower *T*_d_ of *syn*-tbu-PyDPI.

## 3. Materials and Methods

### 3.1. General

All of the solvents and reactants are commercially available and were used without purification. The ^1^H NMR and spectra were recorded on Bruker AVANCZ 500 spectrometers (Bruker, Bremen, Germany) at 298K by utilizing deuterated tetrahydrofuran (THF) as solvent and tetramethylsilane (TMS) as the internal standard. The mass spectra were recorded on a Waters XevoTQ mass spectrometer (Waters, VA, USA). The elemental analysis was performed on FLASH EA1112 elemental analyzer. Absorption spectra were collected using a PERSEE UV-1900 ultraviolet spectrophotometer (Shimadzu, Kyoto, Japan). Fluorescence emission spectra were obtained on a Perkin Elmer LS55 fluorospectrophotometer (North Billerica, MA, USA). Diffraction data were collected on a Rigaku RAXIS-PRID diffractometer using the ω-scan mode with graphite-monochromator Mo•Kα radiation. The structures were solved with direct methods using SHELXTL and refined with full-matrix least-squares on F^2^. In the case of *syn*-tbu-PyDPI, the *tert*-butyl group was found to be disordered on three sites with an occupancy factor of 0.629/0.371 for C48/C48′, C49/C49′ and C50/C50′. Thermal gravimetric analysis (TGA) was measured on a Perkin-Elmer thermal analysis system from 50 °C to 800 °C at a heating rate of 10 K min^−1^ under nitrogen flow rate of 80 mL min^−1^. Differential scanning calorimetry (DSC) was performed on a NETZSCH (DSC-204) unit from 50 °C to 450 °C at a heating rate of 10 K min^−1^ under a nitrogen atmosphere. The ground state geometries were optimized at the B3LYP/6-31G(d,p) level, which is a common method to provide molecular geometries. The HOMO/LUMO distributions are calculated on the basis of optimized ground-state.

### 3.2. Synthesis of Isomers Anti-tbu-PyDPI and Syn-tbu-PyDPI

The starting material 2,7-di-tert-butylpyrene-4,5,9,10-tetraone was prepared according to the literature procedures [40]. A mixture of aniline (4.6 mL, 50.0 mmol), 2,7-di-tert-butylpyrene-4,5,9,10-tetraone (1.87 g, 5.0 mmol), corresponding benzaldehyde (1.3 g, 12.0 mmol), ammonium acetate (3.1 g, 40.0 mmol), and acetic acid (20 mL) was refluxed under nitrogen in an oil bath. After 2 h, the mixture was cooled and filtered. The solid product was washed with an acetic acid/water mixture (1:1, 150 mL). And then, the two isomers were separated by chromatography using CH_2_Cl_2_ as eluent and further purified by sublimation under vacuum.

2,8-*di*-tert-butyl-4,5,10,11-tetraphenyl-4,10-dihydropyreno[4,5-*d*:9,10-*d*’]diimidazole (*anti*-tbu-PyDPI): White powder (yield: 35.0%), ^1^H NMR (500 MHz, THF-*d*_8_, 25 °C, δ): 9.00 (d, *J* = 1.9 Hz, 2H), 7.78–7.67 (m, 14H), 7.57 (d, *J* = 1.9 Hz, 2H), 7.32–7.24 (m, 6H), 1.31 (s, 18H). MALDI-TOF MS (mass *m*/*z*): calcd. for C_50_H_42_N_4_, 698.34; found, 699.5 [M + H^+^]. Anal. Calcd. for C_50_H_42_N_4_: C. 85.93; H. 6.06; N. 8.02; Found: C, 85.74; H, 6.33; N, 7.98.

2,8-*di*-tert-butyl-4,5,11,12-tetraphenyl-4,12-dihydropyreno[4,5-*d*:9,10-*d*’]diimidazole (*syn*-tbu-PyDPI): White powder (yield: 32.5%), ^1^H NMR (500 MHz, THF-*d*_8_, 25 °C, δ): 8.96 (s, 2H), 7.78 (dd, *J* = 6.5, 3.2 Hz, 4H), 7.76–7.71 (m, 6H), 7.69 (dd, *J* = 6.5, 3.2 Hz, 4H), 7.45–7.39 (m, 6H), 7.35 (s, 2H), 1.69 (s, 9H), 0.84 (s, 9H). MALDI-TOF MS (mass *m*/*z*): calcd. for C_50_H_42_N_4_, 698.34; found, 699.5 [M+H^+^]. Anal. Calcd. for C_50_H_42_N_4_: C. 85.93; H. 6.06; N. 8.02; Found: C, 85.88; H, 6.25; N, 7.88.

## 4. Conclusions

Two structural isomers containing pyrene–imidazole skeleton, *anti*-tbu-PyDPI and *syn*-tbu-PyDPI with different molecular symmetries, were conveniently synthesized and successfully separated through a one-pot reaction. The structural characteristics of *anti*/*syn*-tbu-PyDPI upon different symmetries were elucidated unambiguously using ^1^H NMR spectroscopy and further confirmed from single-crystal X-ray measurement. In addition, the similarities with differences of intermolecular interaction and packing mode were accurately characterized by X-ray crystallography. Crystal packing in *anti*-configuration with centrosymmetry strengthened intermolecular interactions displaying higher thermostability. The preliminary fluorescence study shows that the two molecules were good candidates for blue light emitting materials. This work provides an efficient synthesis method and experimental basis for the construction of stable, novel, and advanced structural isomers based on pyrene–imidazole with very convenient accessibility.

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
