# Peer review of "Synthesis and Configurational Character Study of Novel Structural Isomers Based on Pyrene–Imidazole"

_molecules, 2019, doi:10.3390/molecules24122293_

Round 1
Reviewer 1 Report
Manuscript ID: molecules-531189 entitled „Synthesis and Configurational Character Study of Novel Structural Isomers Based on Pyrene-Imidazole” describes preparation and characterization of the structure of new molecule of pyrene-based. The authors have synthesized new compounds using an easy and fast “one-pot” method. The work presents well documented characterization of the structure of new syn-/anti-structural isomers consisting of pyrene and imidazole. The work describes the structure, molecular configurations of crystal and optical and thermal properties of these isomers. Presented in the work, easy to receive and stable isomers, can be good candidates for blue light emitting materials. The article is written carefully and contains only few typographical errors.
The presented results are a continuation of earlier studies in which the authors synthesized syn-/anti-isomers using the same methodology, but used an aldehyde with a different structure [Ref. 29]. However, the described synthesis is a modification of general method Debus‐Radziszewski reaction of the synthesis of imidazoles from a dicarbonyl, an aldehyde, and ammonia. The authors should consider adding this information to the introduction. Additionally, please correct in the 234 line: “tetrahydrofuran (THF) as solvent” into : “deuterated tetrahydrofuran (THF) as solvent”
Author Response
Dear Reviewer,
Thank you for your hard work. We have revised the manuscript carefully according to your kind suggestions. All the revision has been marked in yellow in the manuscript.
The responses to your commons were list at the end of the letter.
If you have any questions, please let me know. I will response as soon as possible. We will try our best to make the paper to fit for publication.
Sincerely yours,
Prof. Dr. Ying Fu
Point 1: The presented results are a continuation of earlier studies in which the authors synthesized syn-/anti-isomers using the same methodology, but used an aldehyde with a different structure [Ref. 29]. However, the described synthesis is a modification of general method Debus-Radziszewski reaction of the synthesis of imidazoles from a dicarbonyl, an aldehyde, and ammonia. The authors should consider adding this information to the introduction
Response 1: Thank you for your comment. We have added this information to the introduction (P. 2, line 83-87) according to your opinion, meanwhile we have also added five references [31-35] to this part for further understanding as follow:
In accordance to previous research [31–33], the “click-like” one-pot reaction is a modification of general method Debus-Radziszewski reaction which has been presented more than 100 years ago by Debus and Radziszewski and has been proved to be a highly efficient multicomponent reaction (MCR) yielding imidazole under mild conditions [34,35].
31. Lindner, J.P. Imidazolium-based polymers via the poly-Radziszewski reaction. Macromolecules 2016, 49, 2046−2053, doi:10.1021/acs.macromol.5b02417.
32. Ma, B.B.; Peng, Y.X.; Zhao, P.C.; Huang, W. cis and trans Isomers distinguished by imidazole N-alkylation after Debus-Radziszewski reaction starting from 2,7-di-tert-butyl-pyrene-4,5,9,10-tetraone. Tetrahedron 2015, 71, 3195−3202, doi:10.1016/j.tet.2015.04.012.
33. Zhu, J.Q.; Li, J.; Wang, X.; Fan, P.; Wang, X.N.; Bian, L.J.; Li, H.Q.; Tian, Y.Q. Facile synthesis of imidazolidines by condensation of aromatic amines with glyoxal and formaldehyde. Curr. Org. Chem. 2012, 16, 1723−1728, doi:10.2174/138527212800840874.
34. Radziszewski, B. Ueber die constitution des lophins und verwandter verbindungen. Ber. Dtsch. Chem. Ges. 1882, 15, 1493−1496, doi:10.1002/cber.18820150207.
35. Debus, H. Ueber die einwirkung des ammoniaks auf glyoxal. Justus Liebigs Ann. Chem. 1858, 107, 199−208, dio:10.1002/jlac.18581070209.
Point 2: Additionally, please correct in the 234 line: “tetrahydrofuran (THF) as solvent” into: “deuterated tetrahydrofuran (THF) as solvent”
Response 2: Thank you for your comment. We have corrected “tetrahydrofuran (THF) as solvent (line 234)” into “deuterated tetrahydrofuran (THF) as solvent” in the revised revision.
Reviewer 2 Report
The manuscript reports the synthesis and characterization of two isomers, viz. syn- and anti- pyrene-imidazole derivatives. Fluorescent properties of the compounds were also studied. This manuscript is easy to read and of interest for the broad readership of the Journal; thus, it can be published after a minor revision:
1. In the Introduction section, abbreviations of the compounds NI-1, NI-2, NI-1-PhTPA and NI-2-PhTPA should be clarified.
2. In the Crystal structure section, some not quite correct expressions are present. For example, there is no high or low degree of centrosymmetry (P. 4, line 119); the molecule can only be centrosymmetric or non-centrosymmetric. The crystal density does not prove higher or lower symmetry of the molecule (P. 4, line 131); its symmetry is uniquely determined from the symmetry of the crystal structure. Moreover, centrosymmetric structure cannot strengthen intermolecular interactions (P. 9, lines 273-274). In fact, both structures (not molecules) are centrosymmetric, because they reveal P21/n space group. Anyway, there is no correlation between symmetry of a crystal and crystal density (amorphous compounds can be more dense than crystalline ones of the same composition). Note that all structural models derived from single crystal XRD experiment have perfect regular crystal packing, and there is no more or less regular crystal packing in this case (P. 9, line 272). Please rewrite the corresponding sentences.
3. It seems that one tert-butyl group of the syn-isomer is disordered. Please mention this in the Experimental section.
4. In my opinion, CH(sp3)···π interactions with H atoms of tert-butyl groups are, in fact, Van der Waals interactions because of low polarity of the corresponding C–H bonds. Furthermore, H···C distances are quite long; thus, I would recommend to exclude discussion of these contacts. Instead, short CH(sp2)···π interactions with phenyl groups at N atoms can be mentioned: in anti-isomer, it is C7···C13(–x, –y, –z) of 3.44 Å (C7···H13(–x, –y, –z) of 2.63 Å), while in syn-isomer, it is C20···C42(1–x, -y, 1-z) of 3.65 Å (C20···H42(1–x, -y, 1-z) of 2.73 Å). These interactions between the neighboring molecules with parallel pyrene moieties are similar for both compounds. Please indicate between which atoms the distances on P. 5 are given. Note that positions of H atoms are unreliable, so it is better to consider C···C/N distances.
5. Please specify excitation wavelengths used to measure the fluorescence spectra.
6. Fig. 5. Please give a vertical scale for the absorption spectra (better to give extinction coefficient values).
7. P. 6, line 201: only one angle of two is listed.
8. In cif file of syn-isomer, color of the crystal is listed “red” instead of “colorless”.
Author Response
Dear Reviewer,
Thank you for your hard work. We have revised the manuscript carefully according to your kind suggestions. All the revision has been marked in yellow in the manuscript.
The responses to your commons were list at the end of the letter.
If you have any questions, please let me know. I will response as soon as possible. We will try our best to make the paper to fit for publication.
Sincerely yours,
Prof. Dr. Ying Fu
Point 1: In the Introduction section, abbreviations of the compounds NI-1, NI-2, NI-1-PhTPA and NI-2-PhTPA should be clarified.
Response 1: Thank you for your comment. We have rewrite the corresponding sentences where the full name of the related compounds are given as follows:
Combined with triphenylamine, a series of donor-accepter molecules based on naphtho[1,2-d]imidazole structural isomers were synthesized, and their different configuration effects on performance of non-doped single-layer fluorescent organic light-emitting diodes were systematically studied. 1H-naphtho[1,2-d]imidazole based compounds emanates similar deep-blue light emission to 3H-naphtho[1,2-d]imidazole analogues but shows much higher efficiency. The author pointed out that the best performance in single-layer devices of 3H-naphtho[1,2-d]imidazole compounds is attributed to their much higher capacity for direct electron-injection from the cathode.
Point 2: In the Crystal structure section, some not quite correct expressions are present. For example, there is no high or low degree of centrosymmetry (P. 4, line 119); the molecule can only be centrosymmetric or non-centrosymmetric. The crystal density does not prove higher or lower symmetry of the molecule (P. 4, line 131); its symmetry is uniquely determined from the symmetry of the crystal structure. Moreover, centrosymmetric structure cannot strengthen intermolecular interactions (P. 9, lines 273-274). In fact, both structures (not molecules) are centrosymmetric, because they reveal P21/n space group. Anyway, there is no correlation between symmetry of a crystal and crystal density (amorphous compounds can be more dense than crystalline ones of the same composition). Note that all structural models derived from single crystal XRD experiment have perfect regular crystal packing, and there is no more or less regular crystal packing in this case (P. 9, line 272). Please rewrite the corresponding sentences.
Response 2: Thank you for your comment.
We have revised the sentence “The numbering guide indicates a high degree of centrosymmetry for anti-tbu-PyDPI……(line 119)” into “The numbering guide indicates a configuration of centrosymmetry for anti-tbu-PyDPI……”.
We have revised the sentence “In contrast, all atomic numbering varied for syn-tbu-PyDPI reflecting its less degree of symmetry……(line 123)” into “In contrast, all atomic numbering varied for syn-tbu-PyDPI reflecting its axisymmetry”.
We have changed the sentence “……which exhibited higher symmetry and more ordered arrangement of the anti-part (line 131)” into “……which exhibited more ordered arrangement of the anti-part”.
We have deleted the sentence “In addition, the density of anti-crystal is higher than that of syn-crystal (Table S1), which further proves higher symmetry and more compact molecular accumulation of anti-configuration (line 132)”.
We have deleted the sentence “As shown in Figure S2, there are twelve CH•••π interactions in one unit cell of anti-tbu-PyDPI which is two more than syn-tbu-PyDPI, further indicating higher symmetry and more compact molecular accumulation of anti-configuration. (line 168)”.
We have revised the sentence “……which was due to its highly rigid structures with high symmetry resulting in stronger and more intermolecular interactions (line 224)” into “which was due to its highly rigid structures and multi-intermolecular interactions”.
We have revised the sentence “Higher molecular symmetry with more compact and more regular crystal packing occurred in anti-configuration resulting in higher thermostability which further manifested that centrosymmetric structure can strengthen intermolecular interactions (line 273)” into “Crystal packing in anti-configuration with centrosymmetry strengthened intermolecular interactions displaying higher thermostability”.
Point 3: It seems that one tert-butyl group of the syn-isomer is disordered. Please mention this in the Experimental section.
Response 3: Thank you for your comment. We have mentioned this information in the experimental section as follow: In the case of syn-tbu-PyDPI, the tert-butyl group was found to be disordered on three sites with an occupancy factor of 0.629/0.371 for C48/C48’, C49/C49’ and C50/C50’.
Point 4: In my opinion, CH(sp3)···π interactions with H atoms of tert-butyl groups are, in fact, Van der Waals interactions because of low polarity of the corresponding C–H bonds. Furthermore, H···C distances are quite long; thus, I would recommend to exclude discussion of these contacts. Instead, short CH(sp2)···π interactions with phenyl groups at N atoms can be mentioned: in anti-isomer, it is C7···C13(–x, –y, –z) of 3.44 Å (C7···H13(–x, –y, –z) of 2.63 Å), while in syn-isomer, it is C20···C42(1–x, -y, 1-z) of 3.65 Å (C20···H42(1–x, -y, 1-z) of 2.73 Å). These interactions between the neighboring molecules with parallel pyrene moieties are similar for both compounds. Please indicate between which atoms the distances on P. 5 are given. Note that positions of H atoms are unreliable, so it is better to consider C···C/N distances.
Response 4: Thank you for your comment. We have excluded the discussion of these contacts which is “For instance, there exist three types of C-H•••π intermolecular interactions in anti-crystal (Figure S2-a). Two types of CH(sp3)•••π interactions (I, II) existed between hydrogen atom of tertiary butyl and the π-electrons of pyrene-imidazole of neighbor molecule with a distance of 3.17 Å and 2.96 Å; the other type of CH(sp2)•••π interaction (III) located between hydrogen atom on phenyl ring attached to the N atom of imidazole and phenyl ring connected to the C atom of imidazole in adjacent molecule with a distance of 2.84 Å. In contrast, there were five types of C-H•••π intermolecular interactions in syn-crystal (Figure S2-b). The first four multiple types of intermolecular CH(sp2)•••π interactions (I, II, III, IV) occur in hydrogen atom on phenyl ring connected to the imidazole and the π clouds of aromatic ring in adjacent molecules such as imidazole, benzene and pyrene with a distance of 3.05 Å, 2.72 Å , 2.81 Å and 2.87 Å respectively; the fifth one was similar to CH•••π interaction (I) in anti-crystal which occurs in hydrogen atom of tertiary butyl and the π-electrons of imidazole of neighbor molecule which is antiparallel alignment (Figure 4) with a distance of 2.87 Å (Figure S2-b).” and added “As shown in Figure S3, in anti-isomer, it is C7···C13(–x, –y, –z) of 3.44 Å (C7···H13(–x, –y, –z) of 2.63 Å), while in syn-isomer, it is C20···C42(1–x, -y, 1-z) of 3.65 Å (C20···H42(1–x, -y, 1-z) of 2.73 Å)” to this part.
Point 5: Please specify excitation wavelengths used to measure the fluorescence spectra.
Response 5: Thank you for your comment. We have added excitation wavelength used to measure the fluorescence spectra as λex = 360 nm.
Point 6: Fig. 5. Please give a vertical scale for the absorption spectra (better to give extinction coefficient values).
Response 6: Thank you for your comment. We have provided a vertical scale for the absorption spectra and gave extinction coefficient values in corresponding part as follow: The molar extinction coefficient of the peak at 304 nm for anti-tbu-PyDPI and the peak at 314 nm for syn-tbu-PyDPI is 8.2 × 104 M-1 cm-1 and 6.3 × 104 M-1 cm-1, respectively, which is large to be of π-π* transitions character.
Point 7: P. 6, line 201: only one angle of two is listed.
Response 7: Thank you for your comment. We added the other angle and revised the sentence “The dihedral angles between phenyl ring attached to the C atom of imidazole and imidazole ring of anti-isomer and syn-isomer are 30°, respectively” into “The dihedral angles between phenyl ring attached to the C atom of imidazole and imidazole ring of anti-isomer and syn-isomer are 30° and 31°, respectively”
Point 8: In cif file of syn-isomer, color of the crystal is listed “red” instead of “colorless”.
Response 8: Thank you for your comment. We have modified the cif file in which color of the crystal is corrected as “colorless” instead of “red”.
